# BIOLOGICAL FACTOR REGULATORY NEURAL NETWORK

## ABSTRACT

Genes are fundamental for analyzing biological systems and many recent works proposed to utilize gene expression for various biological tasks by deep learning models. Despite their promising performance, it is hard for deep neural networks to provide biological insights for humans due to their black-box nature. Recently, some works integrated biological knowledge with neural networks to improve the transparency and performance of their models. However, these methods can only incorporate partial biological knowledge, leading to suboptimal performance. In this paper, we propose the Biological Factor Regulatory Neural Network (BFReg-NN), a generic framework to model relations among biological factors in cell systems. BFReg-NN starts from gene expression data and is capable of merging most existing biological knowledge into the model, including the regulatory relations among genes or proteins (e.g., gene regulatory networks (GRN), protein-protein interaction networks (PPI)) and the hierarchical relations among genes, proteins and pathways (e.g., several genes/proteins are contained in a pathway). Moreover, BFReg-NN also has the ability to provide new biologically meaningful insights because of its white-box characteristics. Experimental results on different gene expression-based tasks verify the superiority of BFReg-NN compared with baselines. Our case studies also show that the key insights found by BFReg-NN are consistent with the biological literature.

## 1 INTRODUCTION

Understanding how cells work is an essential problem in biology, and it is also very important in biomedical areas because of disease phenotype and precision medicine. From a genome-scale view, the whole cell system is modeled by level, starting from DNA, mRNA, and protein to metabolomics, and finally, inferring the phenotype. We define these molecules and molecule sets as biological factors. At each level, the same type of biological factors interact or regulate each other, which determines cell fate, driving the cells to develop, differentiate, and do other activities (Angione, 2019). Thanks to single-cell sequencing technologies, we can obtain gene expression data from the mRNA level, which is fundamental to analyzing the whole cell system. Currently, gene expression data is widely used to identify cell states during cell development, characterize specific tissues or organs, and analyze patient-specific drug responses (Paik et al., 2020).

Many deep learning methods are proposed to utilize gene expression data for predictions, and achieve extraordinary performance in different biological tasks. For instance, gene expression could be treated as a type of input feature to classify cell types, cluster cells and even calculate patient survival time (Erfanian et al., 2021; Huang et al., 2020). Although most deep neural networks (DNNs) model could diagnose cancers with high precision, the original DNNs cannot tell us detailed biological factors/processes which cause cancers. For instance, the regulation between gene PFKL and HIF1A under HEPG2 pathway has a high probability of causing liver cancer (Shoemaker, 2006; Garcia-Alonso et al., 2019).

Recently, some works leverage existing biological knowledge as graphs to represent the relations of biological factors into the prediction models, and significantly improve the prediction accuracy of specific tasks. For example, Rhee et al. (2018) and Chereda et al. (2021) mapped gene expression data into the protein-protein interaction network, and used graph neural networks to predict cancer. Elmarakeby et al. (2021) modeled the relations of gene-pathway and pathway-biological process as

a network, and used a deep neural network to diagnose prostate cancer. Yu et al. (2016); Ma et al. (2018) used the Gene Ontology (GO) knowledgebase to build the neural network architecture, but they are too sketchy to simulate the gene or protein reactions in the cells, and may lead to suboptimal performance. Although they mitigate the black-box issues, they only use partial biological knowledge, and they cannot explore new knowledge from gene expression data.

In this paper, we propose a generic framework, named biological factor regulatory neural network (BFReg-NN), whose goal is to simulate the complex biological processes in a cell system, understand the functions of genes or proteins, and ultimately give insights into the mechanism of larger living systems. Particularly, BFReg-NN is a neural network with the following characteristics. First of all, each neuron is mapped into a biological factor (e.g., a specific gene or protein), and arranged level by level based on the hierarchy of biological concepts, such as genes, proteins, pathways, biological processes, and so on. Secondly, since biological factors regulate each other, edges between neurons (and hyperedges among neurons) are set to reflect the existence of these regulations. In such a manner, edges also model biological meanings. Moreover, two different operations are utilized to simulate the reactions inside/across the layer. Since genes regulate each other and create feedback loops to form cyclic chains of dependencies in gene regulatory network, graph neural network styled operations are suitable to model the "steady state" of genes. It is the same for proteins in PPI. In the layer of pathways, it is a hypergraph where each hyperedge is a pathway including multiple proteins. Accordingly, a hypergraph neural network is used to aggregate and balance the information of each protein. The operations across the layers imitate the material transformation (e.g., genes translate protein), so we adopt deep neural network styled operations to map the relations. Finally, BFReg-NN is flexible to explore new biological knowledge by adding important but not existing edges in the current biological neural network. We illustrate BFReg-NN simulates on genome-scale cell system as an example in Figure 1.

The advantages of our proposed model include: (1) Compared with previous works, BFReg-NN merges with the structural biological knowledge in cell systems, including hierarchical relations (e.g., genes-proteins, proteins-pathways mappings), and regulatory relations among certain factors, such as GRN and PPI. Therefore, it could imitate how different biological factors work inside a cell. (2) The model of BFReg-NN is transparent and interpretable, as each neuron and edge has its corresponding biological meaning. Thus, the learned model weights give evidence of which biological parts are activated and which biological products are generated, leading to the final prediction. (3) By adding new edges between neurons, BFReg-NN not only achieves better performance in downstream tasks, but also has the potential to complete undiscovered biological knowledge. Traditional knowledge completion methods for biological domains (e.g., link prediction by knowledge bases/graphs) suffer from imbalanced data problems (Bonner et al., 2022). BFReg-NN utilizes the gene expression data, which reflects the real cell states, and thus obtains more reliable results.

In the experiment, we show the effectiveness of BFReg-NN on several biological tasks which have different output formats, including missing gene expression value prediction, cell classification and future gene expression value forecasting. We also test the knowledge completing ability of BFReg-NN by the recall of the existing biological knowledge. Further, we do case studies for newly discovered knowledge. The results demonstrate that BFReg-NN provides biologically meaningful insights.

## 2 RELATED WORK

**Gene expression and its applications:** By RNA sequencing, it is easy to obtain gene expression which is a value to represent the amount of gene transcripts from a DNA fragment (Eberwine et al., 2014). It has been used in a variety of biological applications, including single-cell analysis (Yu et al., 2022; Zhou et al., 2022), disease diagnosis (Xing et al., 2022) and drug discovery (Pham et al., 2021). But most of these models lack transparency and ignore the existing biological knowledge.

**Knowledge graph enhanced downstream tasks:** The emergence of knowledge bases/graphs has led to enhancing the performance in many fields of computer science, such as computer vision and natural language processing (Ren et al., 2021; Hao et al., 2021; Liu et al., 2021). Similarly, knowledge graphs also have been widely used for specific biological tasks such as cancer diagnosis in recent years (Elmarakeby et al., 2021; Rhee et al., 2018). OntoProtein (Zhang et al., 2022) embedded the gene ontology knowledge in pre-training to improve the performance of several pro-

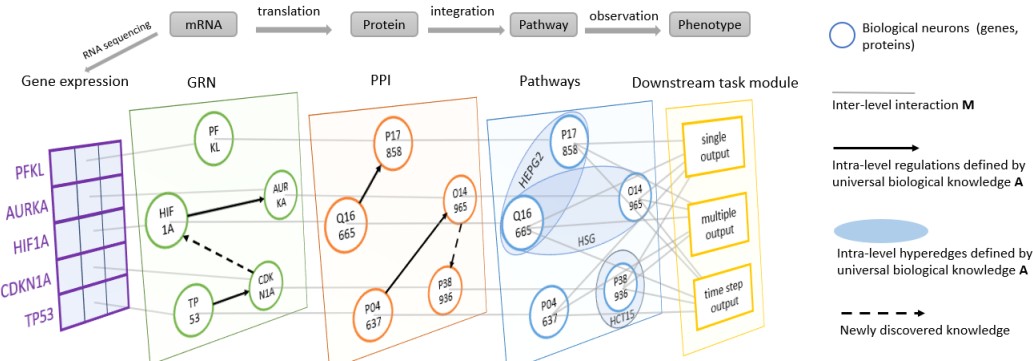

Figure 1: The pipeline of BFReg-NN. We build the hierarchical biological network inspired by the cell system, modeling several levels to separate mRNA, protein, pathway and phenotype, and simulating biological factor interactions both at the intra-level and inter-level. We obtain the gene expression value from RNA sequencing as input, to predict the property of the phenotype as output. Taking the cancer dataset NCI-60 as an example. We know that the regulatory activations start from gene HIF1A and TP53 to gene AURKA and CDKN1A at the gene level (Garcia-Alonso et al., 2019). And at the protein level, the translated proteins Q16665 and P04637 stimulate P17858 and O14965 respectively (Türei et al., 2016). We mark those relations as existing biological knowledge by the black solid lines. Multiple genes with their products consist of pathways to drive cells to different types. For example, PFKL and HIF1A are activated in the HEPG2 cell line pathway, which leads to a type of liver cancer (Shoemaker, 2006). Besides, new regulatory relations would be identified at each level shown as dotted lines. After training, we learn the good representations of biological factors, and employ them for different downstream tasks with various output formats.

tein downstream tasks. But it limited the interpretation ability because of transforming knowledge into embeddings. Elmarakeby et al. (2021) modeled the relations of gene-pathway and pathway-biological process as a neural network to diagnose prostate cancer. Chereda et al. (2021) assigned genes to the PPI network to predict breast cancer, and then explained the important genes by Layer-wise Relevance Propagation. Although they mitigate the black-box issues, they all use a small part of biological knowledge, and they cannot explore new knowledge from gene expression data.

**Knowledge complement:** Our work is related to knowledge complement (Yao et al., 2020; Goel et al., 2020). In particular, lots of work on biological knowledge complement has been done recently (Yu et al., 2021; Zitnik et al., 2018). Much attention is paid to predicting the relation between specific biological factors, which can be regarded as a link prediction problem. For example, Mohamed et al. (2020) predicted drug-target interaction (DTI) by learning representations of drugs and targets from the KEGG database. Hamilton et al. (2018) discovered new drugs for diseases by embedding the drug-gene-disease database in multiple relations. However, existing biological knowledge is collected in imbalance, which causes little biological meaningful predictions (Bonner et al., 2022). In our work, we expect discovered knowledge from gene expression data, which can interpret the cell state by biological factor interactions and avoid imbalanced situations.

## 3 BIOLOGICAL BACKGROUND

The biological system is modeled by a complex network that connects many biologically relevant entities to work together to perform one or more particular functions. It could be at the organ/tissue scale, such as the nervous system, or the integumentary system. On the micro/nanoscopic scale, examples include cells, organelles, and so on. In this work, we focus on the simulation of the biological system in a cell at a genome-scale.

Thanks to the development of sequencing technologies, it is easy and cheap to obtain an amount of gene expression data to build genome-scale analysis. Starting from genes, we model the cell genome-scale system by single-omic level (intra-level) regulations and different omic level (inter-level) mappings (Angione, 2019). Among different omic levels, defined as inter-level in BFReg-

NN, genes are first transcribed from DNA and then translate to proteins. Then, pathways integrate individual genes or protein products to perform certain cell functions. To model single-omic level regulations, we define the intra-level interactions. Genes are regulated by each other, known as GRN, which means the gene expression values are governed to be inhabited or activated by association molecule products, such as RNA. Proteins interact with each other in a similar way, inhabiting, activating, or combining with others to influence the expression values (or protein abundances) in cells, called PPI. Notice that in general, the relations among factors are universal for all the cells, thus the biological knowledge could adapt to different types of cells. But sometimes the biological factor interaction would rewire in specific cells, and thus the extra knowledge discovered and verified is also important (Lynch et al., 2011).

## 4 METHOD

Assume we have gene expression data $\mathbf{x} \in \mathbf{R}^n$ show the state of a cell, where $n$ is the number of genes. Besides, we also define the neural network architecture of BFReg-NN according to structural biological knowledge with the intra-level regulations $\mathcal{A}$ and the inter-level mappings $\mathcal{M}$. After training BFReg-NN, we obtain the hidden embedding $\mathbf{H}_i = \text{BFReg-NN}(\mathbf{x}, \mathcal{A}, \mathcal{M})$ for each gene $i$, and do predictions for downstream tasks.

The output format of predictions could be a single value $y = f(\mathbf{H})$, or multiple values $\bar{\mathbf{x}} = f(\mathbf{H})$. Further, we also handle the time-series output. Given the gene expressions of cells at the $t_0$ time, we aim to predict the gene expressions $\hat{\mathbf{X}} \in \mathbf{R}^{n \times T} = f(\mathbf{H})$ in the following $t_1, \ldots, T$ time steps. Finally, BFReg-NN could explore new insights by adding more edges $\mathcal{A}'$.

In the following sections, we first extract $\mathcal{A}$ and $\mathcal{M}$ from the existing biological databases/graphs. Then, we propose two versions of BFReg-NN. Finally, we introduce how to apply BFReg-NN to different downstream tasks.

### 4.1 BIOLOGICAL KNOWLEDGE DATABASES/GRAPHS

Based on the existing biological knowledge, we first divide biological factors into different levels, $\mathcal{L} = \{\text{Gene}, \text{Protein}, \text{Pathway}, \ldots\}$. At each level, regulatory relations between factors in the intra-level are formulated as a matrix set $\mathcal{A} = \{\mathbf{A}_{\text{Gene}}, \mathbf{A}_{\text{Protein}}, \mathbf{A}_{\text{Pathway}}, \ldots, \mathbf{A}_L\}$, where $\mathbf{A}_{\text{Gene}}$ could be gene relations and defined by GRN, and $\mathbf{A}_{\text{Protein}}$ is decided by PPI. $\mathbf{A}_{\text{Pathway}}$ is a little special because it is a hypergraph, where each edge could connect more than two nodes, called the hyperedge. The proteins in a hyperedge propagate the information to influence each other directly. The values in $\mathbf{A}_l$ are binary to represent the existence of relations. We also identify the binary mapping matrixes $\mathcal{M} = \{\mathbf{M}_1, \mathbf{M}_2, \ldots, \mathbf{M}_{L-1}\}$ from level $l$ to its upper level $l + 1$ as the inter-level interaction. The dimension of $\mathbf{M}_l$ is dependent on the factor numbers of the two levels. We set $\mathbf{M}_1$ to map between genes and proteins, and $\mathbf{M}_2$ as a direct mapping between protein-level and pathway-level. $\mathcal{A}$ and $\mathcal{M}$ both decide the architecture of the neural network, including neurons and links between neurons.

### 4.2 BASIC BFREG-NN MODEL

Given the gene expression of a cell $\mathbf{x} \in \mathbf{R}^n$, we first utilize an embedding layer to encode each gene independently, where $\mathbf{H}_i^{0,0} = \text{Emb}(\mathbf{x}_i)$. The embedding layer let genes share a same parameter MLP layer to obtain the gene expression representation. As the gene expression is in a float form with a little measurement error, the embedding layer is utilized to reduce this error and thus enhance the performance of the model. Then we obtain $\mathbf{H}^{0,0} \in \mathbf{R}^{n \times d}$ as the input to neurons at the first level, i.e., gene neurons.

We have $\mathbf{A}_l$ as intra-level relations between neurons at level $l$. Inspired by graph neural networks, we use the message passing mechanism to update $\mathbf{H}^{l,k}$, which is the hidden representation of the neuron in the $k$-hop at level $l$. The formulation is:

$$\mathbf{H}_i^{l,k+1} = \text{update}\left(\sum_{j \in \mathbf{A}_l(i)} \text{message}\left(\mathbf{H}_i^{l,k}, \mathbf{H}_j^{l,k}\right), \mathbf{H}_i^{l,k}\right). \tag{1}$$

The message function is to generate the message from neuron $j$ to neuron $i$, where $\mathbf{A}_l(i)$ decides which neurons are neighbors for neuron $i$. Then the update function is to update the embedding

of neuron $i$ using the obtained messages and the previously hidden embedding. $k \in [0, K-1]$ is the number of hops to determine that neuron $i$ is influenced by other neurons in the $K$-hop neighborhood. In detail, we utilize GAT-styled (graph attention) (Vaswani et al., 2017) to compute the attention weight for each message, and weighted sum up all the messages to update. As for operations in hypergraphs, we apply HGNN-styled (hypergraph neural network) (Bai et al., 2021) message and update functions. Batch normalization is conducted at the end of each level.

Due to multiple level interactions in biological systems, the embedding is also learned level by level. Since $\mathbf{M}_l$ is inter-level relations between level $l$ and $l+1$, we utilize the masked neural network to update the initial representation $\mathbf{H}^{l+1,0}$ at level $l+1$:

$$\mathbf{H}^{l+1,0} = \text{activation}\left( (\mathbf{M}_l \odot \mathbf{W}_l) \mathbf{H}^{l,K} + \mathbf{b}_l \right) \tag{2}$$

$\mathbf{H}^{l,K}$ is the output after batch normalization. The element-wise multiplication $\mathbf{M}_l \odot \mathbf{W}^l$ ensures that non-existing relations are not used for updating. In fact, the important biological factors could be ranked by the weighted matrix $\mathbf{W}^l$ calculated between levels. $\mathbf{W}_l$ and $\mathbf{b}_l$ are learnable parameters.

### 4.3 ENHANCED BFREG-NN MODEL

Here the enhanced version is introduced by adding new edges in $\mathcal{A} = \{\mathbf{A}_{\text{Gene}}, \mathbf{A}_{\text{Protein}}, \mathbf{A}_{\text{Pathway}}, ...\}$. It can explore new biological insights and improve the performance simultaneously.

Existing biological knowledge is detected by biological technology to reflect the implicit relations among factors. However, some knowledge is still hard to be discovered due to technological limitations and rewiring phenomenons in individual cells. Therefore, we spilt the interaction into two types. One is the universal regulation, supported by existing knowledge $\mathcal{A}$. The other is local interaction, inferring the new biological knowledge or rewiring in individual cells, but now hidden in the non-existent edges of $\mathcal{A}$.

Instead of a binary matrix $\mathbf{A}_l$ used in the basic model, we use $\mathbf{A}_l$ to constrain the learnable matrix $\mathbf{A}'_l$ to discover new knowledge. As universal regulations are verified by biological methods, we use them the same as the basic model. For non-existent edges, we reweigh it by a small value $0 \leq \alpha < 1$ due to it being less convincing. Thus, the edge intensity based on two types of knowledge is modified as:

$$\mathbf{A}'_l = \begin{cases} w_{ij}^l, & \text{universal regulation between } i \text{ and } j \text{ and it already exists in } \mathbf{A}_l \\ \alpha w_{ij}^l, & \text{local interaction between } i \text{ and } j \text{ but it is ignored in } \mathbf{A}_l \end{cases} \tag{3}$$

where $w_{ij}^l = \sigma(\text{MLP}(\text{concat}[\mathbf{H}_i^l, \mathbf{H}_j^l]))$. In the enhanced model, we not only learn the neuron embeddings, but also utilize these embeddings with an MLP transformation to infer the intensity of the hidden interaction between neuron $i$ and $j$. Then we update the representations by

$$\mathbf{H}^{l,k+1} = \sigma\left( \text{MLP}\left( \mathbf{A}'_l \mathbf{H}^{l,k} \right) \right). \tag{4}$$

After the model is trained to converge, we obtain the learned weights for non-existence edges, which provides insights for new knowledge and the rewiring phenomenon. We sort the weights $w_{ij}^l$ to identify the candidates which deserve verification by biological experiments. Here we use a simple edge weight modification method instead of gated edges implemented by the gumble-softmax function or advanced graph structure learning algorithms (Zheng et al., 2020; Jin et al., 2020). The reason is that the biological knowledge is not sparse, and even dense in some core genes. Thus, it does not satisfy the sparse and low-rank requirements. Our simple method does not add much extra computation cost while achieving great effectiveness in biological tasks.

### 4.4 DOWNSTREAM TASKS

After we obtain the final embedding $\mathbf{H}^{L,K}$, we could conduct different types of downstream tasks, whose output format could be one-dimensional, multiple-dimensional, or time-series. Here we illustrate them by three specific tasks, missing gene expression value prediction, future gene expression value forecasting, and cell classification.

**Missing gene expression value prediction:** Suppose we have a gene expression vector $\mathbf{x} \in \mathbf{R}^n$ for $n$ genes, measured by the single-cell sequencing technology. Some parts in the vector are zeros

or in a low value because of dropout events (Gong et al., 2018), causing biases among cells. So we aim to accurately recover all the missing values and obtain a new vector $\bar{\mathbf{x}}$. Since the gene expression is dependent on a certain cell, we merge gene representations. The new gene value vector is generated directly by $\bar{\mathbf{x}} = \text{MLP}(\mathbf{H}^{L,K}) = \text{MLP}(\text{BFReg-NN}(\mathbf{x}))$. We minimize the mean squared loss between predicted values and ground truth for all the non-zero elements in $\mathbf{x}$. Finally, the missing values are imputed with the predicted values.

**Cell classification:** This task is to classify the cell types using gene expression values $\mathbf{x}$, which is important to determine the situation of tissues or patients. Because cell type is the observable result of a biological system, we simulate each gene to pass the multiple levels of transformation to infer the property of the cell. The prediction is computed by $\hat{y} = \text{MLP}(\mathbf{H}^{L,K}) = \text{MLP}(\text{BFReg-NN}(\mathbf{x}))$. It is a multi-class classification task and we employ a cross-entropy loss.

**Future gene expression value forecasting:** The single-cell data is efficient to analyze the cell response under different drugs. However, it is expensive to collect the data at different time steps and draw the development trend of cells. Thus, we model the cell response by future gene expression value prediction. In other words, given the gene expression data $\mathbf{x} \in \mathbf{R}^n$ at $t_0$, we aim to forecast the gene expression data $\hat{\mathbf{X}} \in \mathbf{R}^{n \times T}$ in the following $t_1, \ldots, T$ time steps. We use two backbones, MLP and LSTM. MLP predicts the gene expression in the future time steps simultaneously, and the equation is $\hat{\mathbf{X}} = \text{MLP}(\text{BFReg-NN}(\mathbf{x}))$. LSTM models the dynamic values step by step, where it takes the last time output as the next step input, and the equation is $\hat{\mathbf{x}}^t = \text{LSTM}(\text{BFReg-NN}(\hat{\mathbf{x}}^{t-1}))$. We employ MSE as the loss function.

## 5 EXPERIMENTS

### 5.1 SETUP

Here we introduce the experimental setting for downstream tasks, including datasets, neural network architecture, and baselines. Training details are presented in Appendix. The code and toy datasets are attached in the supplementary material.

**Missing gene expression value prediction:** In this task, we collect 10 cell lines (BT20, HS578T, LNCAP, A549, MCF7, MCF10A, MDAMB231, PC3, SKBR3 and A375) from the L1000 dataset. Because of the limitation of sequencing technology, the L1000 dataset does not have the golden standard values for non-landmark genes. Thus, we employ the pre-processed data based on Qiu et al. (2020). Besides, we also ignore the time point information in cell lines. In each cell line, we have 1482 cell samples and 714 genes. We also collect 541 and 2305 existing knowledge edges for GRN and PPI respectively. We randomly mask the gene expression value with a 60% probability, and predict these missing values. The cell samples are randomly split by 60/20/20% as training/validation/test data. The evaluation metric is mean square error (MSE), which means the smaller the better.

**Cell classification:** We gather 5 datasets to predict cell types. The first one is GSE756888 (breast cancer) collected from Chung et al. (2017). The following four datasets are about organs, including muscle, diaphragm, lung and trachea, obtained from Consortium et al. (2018). We pre-process the data by deleting the cells and genes which have a larger ratio of zeros. The remaining data are summarized in Table 6. We also split the data same as the missing value task, and evaluate the results by macro AUC.

**Future gene expression value forecasting:** For future value prediction, we choose the breast cancer dataset from 2019 Dream Challenge (Gabor et al., 2021). There are 44 cell lines and 6 treatments. Different treatments mean using various drugs on cells to test the drug effects. Since the data are quantified at the proteomic level, we have 37 biomarkers as genes and 38 related PPIs as existing knowledge, which is a small protein-protein interactions graph where nodes are phosphorylated genes. The detailed process to obtain dynamic gene expression data is: all the cells are first treated and then divided into several groups. At regular intervals, a group of cells is killed and measured their gene expression. To avoid noise caused by this measurement on each individual cell, we calculate the median for each gene at every time step. Different from the previous two tasks, we use 5 treatments as training data, and the remaining one as test data. The 5-fold cross validation is done. We use MSE and Pearson correlation coefficient (PCC) to evaluate average performance.

Table 1: Results on missing gene expression value prediction (evaluated by MSE ↓).

| Knowledge | Models | BT20 | HS578T | LNCAP | A549 | MCF7 | MCF10A | MDAMB231 | PC3 | SKBR3 | A375 |
|---|---|---|---|---|---|---|---|---|---|---|---|
| no knowledge | MLP | **0.314** | 0.344 | 0.344 | 0.320 | 0.326 | 0.337 | 0.325 | 0.378 | 0.317 | 0.336 |
| | transformer | 0.396 | 0.377 | 0.406 | 0.383 | 0.396 | 0.399 | 0.418 | 0.402 | 0.387 | 0.407 |
| | Gated NN | 0.709 | 0.681 | 0.644 | 0.533 | 0.647 | 0.700 | 0.667 | 0.722 | 0.594 | 0.729 |
| co-expression | GCN | 0.715 | 0.714 | 0.722 | 0.653 | 0.688 | 0.675 | 0.707 | 0.731 | 0.677 | 0.702 |
| | GAT | 0.719 | 0.715 | 0.723 | 0.655 | 0.693 | 0.669 | 0.720 | 0.721 | 0.676 | 0.716 |
| existing knowledge | MLP+N2V | 0.351 | 0.378 | 0.380 | 0.349 | 0.360 | 0.350 | 0.358 | 0.392 | 0.343 | 0.354 |
| | GCN | 0.331 | 0.329 | 0.330 | 0.304 | 0.340 | 0.313 | 0.319 | 0.338 | 0.331 | 0.317 |
| | GAT | 0.324 | 0.331 | 0.332 | 0.308 | 0.321 | 0.314 | **0.317** | 0.347 | 0.317 | 0.315 |
| | P-NET | 0.349 | 0.335 | 0.340 | 0.340 | 0.321 | 0.352 | 0.336 | 0.354 | 0.322 | 0.331 |
| | BFReg-NN(Basic) | 0.318 | 0.336 | 0.332 | 0.329 | **0.308** | 0.328 | 0.328 | 0.344 | 0.318 | 0.328 |
| | BFReg-NN(Enhanced) | 0.318 | **0.317** | **0.320** | **0.297** | **0.308** | **0.306** | 0.320 | **0.335** | **0.307** | **0.308** |

Table 2: Results on cell classification (evaluated by macro AUC ↑).

| Knowledge | Models | GSE | muscle | diaphragm | lung | trachea |
|---|---|---|---|---|---|---|
| no prior knowledge | MLP | 0.9122 | 0.8586 | 0.7881 | 0.8545 | 0.9387 |
| | transformer | 0.9476 | 0.8785 | **0.8717** | 0.8900 | 0.9242 |
| | Gated NN | 0.7910 | 0.7784 | 0.7110 | 0.8240 | 0.9120 |
| co-expression | GCN | 0.6916 | 0.6424 | 0.5945 | 0.5822 | 0.6612 |
| | GAT | 0.7790 | 0.6882 | 0.6305 | 0.6312 | 0.6294 |
| existing knowledge | MLP+N2V | 0.9064 | 0.8772 | 0.7501 | 0.8180 | 0.9404 |
| | GCN | 0.9285 | 0.8449 | 0.7896 | 0.8581 | 0.9421 |
| | GAT | 0.9255 | 0.8470 | 0.8039 | 0.8246 | 0.9409 |
| | P-NET | 0.9052 | 0.8654 | 0.7973 | 0.8425 | 0.9332 |
| | BFReg-NN(Basic) | 0.9476 | 0.8808 | 0.8420 | 0.8808 | 0.9376 |
| | BFReg-NN(Enhanced) | **0.9693** | **0.8884** | 0.8509 | **0.8903** | **0.9446** |

**Neural network architecture:** The architecture is totally defined by existing knowledge. From open-source databases, we select Dorothea (Garcia-Alonso et al., 2019) as GRN construction, and Omnipath (Türei et al., 2016) as PPI. As the pathway is specific in different cells, we collect pathways for each dataset by Enrich (Chen et al., 2013), which provides knowledge of the hyperedge connection among genes and proteins, including LINCS_L1000_Ligand_Perturbations for missing value task and WikiPathways_2019_MOUSE(HUMAN) for cell classification task. For future value forecasting, we utilize its corresponding relations in the dataset.

**Baselines:** We divide baselines into three types: (1) There is no prior knowledge integrated into the model, and the input is only gene expression data. **MLP** is the simplest model for static prediction. **LSTM** is the typical RNN-based model for dynamic value prediction. We also select more complex models, **Transformer** (Vaswani et al., 2017) and **GatedNN** as competitors. GatedNN is an improved version of deep neural networks, where edges between neurons are gated by the gumble-softmax function so the network could be sparse. (2) The second type of baselines is to first learn a gene co-expression matrix as knowledge and then do the prediction. The co-expression matrix identifies which genes have a tendency to show a coordinated expression pattern, so it can be built by the similarity between gene expressions. We follow the process described in MLA-GNN (Xing et al., 2022) to learn the knowledge and then apply two classical graph neural networks, **GCN** (Kipf & Welling, 2017) and **GAT** (Veličković et al., 2018), to aggregate the information and do the prediction. (3) We also compare the models using the existing biological knowledge. This knowledge could be graphs, such as GRN or PPI. **MLP+N2V** used node2vec (Grover & Leskovec, 2016) to learn node embeddings of the prior graph, combine them with gene expression data and employ MLP to predict. **GCN** and **GAT** directly utilized the prior graph. As there are different prior graphs (GRN or PPI), we report the best results for GCN and GAT with the most suitable graph. **P-NET** (Elmarakeby et al., 2021) utilized protein-pathway relations as prior knowledge and apply DNNs to predict. For our proposed model **BFreg-NN**, we test basic and enhanced versions.

## 5.2 EXPERIMENTAL RESULTS

**Static task results:** We report the main results for missing value prediction and cell classification, summarized in Table 1 and Table 2 respectively. Firstly, in general, compared to no prior knowledge methods, biological knowledge improves the performance of models effectively, which means that modeling specific meanings for neural networks could improve the performance in biological tasks. Although Gated NN includes the gumble-softmax function which also is able to predict edges in a

Table 3: Results on future gene expression value forecasting (evaluated by MSE↓ and PCC↑).

| Knowledge | Models | MSE | PCC |
|---|---|---|---|
| no prior knowledge | MLP | $0.0761 \pm 0.0038$ | $0.9704 \pm 0.0018$ |
| | LSTM | $0.0822 \pm 0.0028$ | $0.8888 \pm 0.0034$ |
| | transformer | $0.0743 \pm 0.0013$ | $0.9717 \pm 0.0007$ |
| | Gated NN | $0.0743 \pm 0.0026$ | $0.9717 \pm 0.0007$ |
| co-expression | GCN | $0.3838 \pm 0.0090$ | $0.8293 \pm 0.0036$ |
| | GAT | $0.3447 \pm 0.0236$ | $0.8455 \pm 0.0114$ |
| existing knowledge | MLP+N2V | $0.0760 \pm 0.0043$ | $0.9706 \pm 0.0018$ |
| | GCN | $0.0920 \pm 0.0053$ | $0.9646 \pm 0.0020$ |
| | GAT | $0.0869 \pm 0.0044$ | $0.9661 \pm 0.0017$ |
| | BFReg-NN (MLP, Basic) | $0.0732 \pm 0.0026$ | $0.9719 \pm 0.0009$ |
| | BFReg-NN (MLP, Enhanced) | $\mathbf{0.0724} \pm 0.0023$ | $\mathbf{0.9724} \pm 0.0007$ |
| | BFReg-NN (LSTM, Basic) | $0.0825 \pm 0.0029$ | $0.8906 \pm 0.0041$ |
| | BFReg-NN (LSTM, Enhanced) | $0.0819 \pm 0.0027$ | $0.8907 \pm 0.0049$ |

Table 4: Ablation study results on cell classification (evaluated by macro AUC ↑).

| Models | Knowledge | muscle | diaphragm | lung | trachea |
|---|---|---|---|---|---|
| Basic | GRN | 0.8165 | 0.7564 | 0.8230 | 0.9376 |
| | GRN&PPI | 0.8798 | 0.8219 | 0.8784 | 0.9241 |
| | GRN&PPI&Pathway | 0.8808 | 0.8420 | 0.8808 | 0.9321 |
| Enhanced | GRN | 0.8165 | 0.7873 | 0.8373 | **0.9446** |
| | GRN&PPI | 0.8807 | 0.8417 | 0.8892 | 0.9297 |
| | GRN&PPI&Pathway | **0.8884** | **0.8509** | **0.8903** | 0.9360 |
| | No Intra-level | 0.8799 | 0.8382 | 0.8809 | 0.9235 |
| | No Inter-level | 0.8790 | 0.8058 | 0.8268 | 0.9391 |

probability, it is hard to generate the existing biological knowledge. Besides, co-expression knowledge does not show outstanding performance because it is extracted from a small size of data and thus brings the noise. Some complex model also achieves promising results, such as Transformer, which indicates that building dense biological relations may be needed for the cell samples. Secondly, BFReg-NN also outperforms the baselines using the existing knowledge. MLP+N2V fails in prediction because it uses the knowledge by embedding nodes implicitly, rather than in an explicit architecture. Compared with GCN, GAT and P-NET, the basic BFReg-NN could merge hierarchical knowledge from different levels of the biological system. Further, the enhanced version learns important undiscovered knowledge from the gene expression data. Overall, the enhanced BFReg-NN achieves the best results in most cases.

**Dynamic task results:** Results for future value prediction are presented in Table 3. Compared with directly employing MLP/LSTM to forecast future values, BFReg-NN with MLP/LSTM could improve the quality of embeddings and achieve better performance. GCN and GAT with co-expression matrix perform badly due to the small dataset. When using existing knowledge, MLP+N2V, GCN and GAT still cannot obtain a good result because the knowledge is incomplete. For example, there are several isolated nodes in the dataset, and these methods fail in updating the embeddings of these nodes. Since transformer and Gated NN could merge the gene information densely, they obtain better results. Finally, the enhanced BFReg-NN could add the extra discovered knowledge to mitigate the problem of incomplete prior knowledge, and learn the embeddings with a suitable architecture. Therefore, it can reach the best performance.

## 5.3 ABLATION STUDY

In this section, we provide a brief description of the effectiveness of each part in BFReg-NN. The ablation analysis for the cell classification task is shown in Table 4. The accuracy is gradually improved when adding a higher level of knowledge to the model in most cases. In trachea dataset, GRN shows the best performance which indicates PPI may lead to an overfitting problem in this organ. In addition, we present the results of *no intra-level* and *no inter-level* situations. *No intra-level* version removes PPI and GRN knowledge and maintains the hierarchical network; and *no inter-level* merges the PPI and GRN into a large graph and deletes the hierarchical structure. The results show that hierarchical structure is more important than simply linking the factors together, which obeys the phenomenon that the biological factors in a cell are produced step by step.

Table 5: Ablation study results on missing value prediction (evaluated by MSE ↓).

| Knowledge | BT20 | HS578T | LNCAP | A549 | MCF7 | MCF10A | MDAMB231 | PC3 | SKBR3 | A375 |
|---|---|---|---|---|---|---|---|---|---|---|
| GRN | **0.318** | **0.317** | **0.320** | **0.297** | **0.308** | **0.306** | **0.320** | 0.335 | **0.307** | **0.308** |
| GRN&PPI | 0.343 | 0.342 | 0.338 | 0.326 | 0.339 | 0.333 | 0.343 | 0.357 | 0.338 | 0.344 |
| GRN&Pathway | 0.335 | 0.335 | 0.326 | 0.314 | 0.324 | 0.319 | 0.331 | **0.332** | 0.327 | 0.325 |

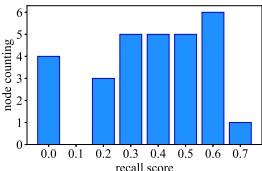
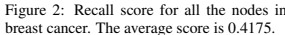

Figure 2: Recall score for all the nodes in breast cancer. The average score is 0.4175.

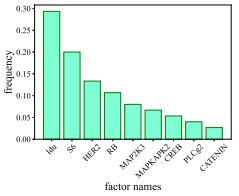

Figure 3: Frequency of newly discovered interactions including p53.

In addition, we could choose different levels from the biological system as layers for BFReg-NN to adapt to different tasks. For example, the cell classification task aims to capture the cell state involving a variety of factors in a cell, and thus it needs multiple levels of biological knowledge to build the cell phenotype. In contrast, shown in Table 5, the missing value task focuses on gene expression value, which means single gene level knowledge is enough while protein or pathway level knowledge may bring the noise to predictions.

### 5.4 Knowledge completion by BFReg-NN

To demonstrate the ability of BFReg-NN to complete the knowledge, we test it on the future value forecasting task with breast cancer to evaluate the knowledge recall score. For each gene in the dataset, we remove its related edges to make it an isolated node, and then conduct the model to recover the knowledge. We run BFReg-NN several times to avoid randomness. In each run, the discovered edges come out with weights, and are ranked to select the top-k list. The frequency of an edge is computed by (the times that the edge is in the top-k list)/(the total runs). We select the top-20 interaction pairs with the highest frequency as discovered knowledge. The results are shown in Figure 2, where BFReg-NN achieves an average 0.4172 recall score for all the nodes. We also take p.p53 gene as example. The detailed recalled edges are shown in Figure D.1. We found all the existing edges for p.p53 in the top-20 list. Overall, we notice that most of the edges are recalled by our model, while BFReg-NN also adds knowledge that does not appear in the existing database.

In recent breast cancer research, some new knowledge is discovered by biological methods, which is also found in our proposed BFReg-NN. We take the gene p53, a well-known tumor suppressor, as an example to verify the biological meaning of discovered knowledge. We show the top-10 frequent interaction pairs including p53 in Figure 3. And then we find the cues in recent biological literature. Iododeoxyuridine (Idu) is the most frequent marker, which identifies cell phases, indicating whether the cell division is continued in breast cancer (Behbehani et al., 2012; Gabor et al., 2021). S6 takes the second place, which is regulated in mTOR signaling by p53, and the level of S6 increases when p53 is insufficiency (Luo et al., 2021). Besides, p53 also regulates the expression level of Her2, where the interaction frequency is about 0.15 in our prediction. As a kind of Her2-positive cancer, breast cancer could be treated by inhibiting p53 (Fedorova et al., 2020). After that, the regulatory relation between p53 and RB is in the existing biological knowledge dataset.

## 6 Conclusion

In this paper, we propose a generic framework BFReg-NN, which offers biological meanings to neurons and links between neurons, and imitates the whole cell system as a neural network at both intra-level and inter-level. We apply BFReg-NN to different downstream tasks and our experimental results demonstrate that BFReg-NN consistently outperforms baselines and discovers new biological meaningful insights. BFReg-NN provides a novel paradigm to merge cell sequencing data and biological knowledge, and it could be extended to more types of genomics data to simulate and understand complex biological systems in the future.

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

# A    APPENDIX - EXPERIMENTAL SETUP

Table 6: statistics for cell classification datasets

| Dataset | #Sample | #Gene | GRN knowledge | PPI knowledge | Pathways | #Label |
|---------|---------|-------|---------------|---------------|----------|--------|
| GSE | 4658 | 418 | 172 | 136 | 89 | 8 |
| lung | 1642 | 382 | 148 | 332 | 121 | 11 |
| muscle | 1051 | 401 | 317 | 401 | 119 | 6 |
| trachea | 3587 | 326 | 109 | 207 | 110 | 5 |
| diaphragm | 853 | 355 | 239 | 299 | 115 | 5 |

Table 7: Hyperparameter setting

| Task | Learning rate | Max epoch | Dimension of hidden embeddings | |
|------|---------------|-----------|---------------|-----------|
| | | | BFReg-NN module | Prediction module |
| missing value prediction | 1e-3 | 200 | 4 | 1024 |
| cell classification | 5e-4 | 200 | 4 | 256 |
| future value forecasting | 1e-4 | 2000 | 16 | 512 |

Table 8: The setting of $\alpha$ in different tasks

| | | | | | Missing value prediction | | | | | |
|---|---|---|---|---|---|---|---|---|---|---|
| Knowledge | $\alpha$ set | BT20 | HS578T | LNCAP | A375 | A549 | MCF7 | MCF10A | MDAMB231 | PC3 | SKBR3 |
| GRN | $\alpha_1$ | 1e-2 | 1e-4 | 1e-5 | 1e-4 | 1e-3 | 0 | 1e-3 | 5e-4 | 1e-3 | 5e-5 |
| GRN&PPI | $\alpha_1$ | 1e-4 | 0 | 5e-4 | 0 | 1e-4 | 5e-5 | 5e-5 | 5e-5 | 5e-4 | 5e-4 |
| | $\alpha_2$ | 0 | 0 | 0 | 0 | 0 | 0 | 0 | 0 | 0 | 0 |

| | | Cell classification | | | | | Future value forecasting | |
|---|---|---|---|---|---|---|---|---|
| Knowledge | $\alpha$ set | GSE | muscle | diaphragm | lung | trachea | Backbone | $\alpha$ set |
| GRN | $\alpha_1$ | 1e-5 | 1e-5 | 5e-5 | 5e-5 | 5e-4 | MLP | 1e-2 |
| GRN&PPI | $\alpha_1$ | 5e-4 | 1e-5 | 1e-5 | 1e-4 | 1e-4 | LSTM | 5e-3 |
| | $\alpha_2$ | 1e-5 | 0 | 5e-5 | 0 | 1e-5 | | |

## A.1    TRAINING DETAILS

Data statistics are summarized in Table 6 for the cell classification task. For optimizers, we use Adam for all the models. The hyperparameters are summarized in Table 7. We set a small dimension of hidden embeddings to avoid overfitting. For the prediction module, we set the number of hidden units with the best value obtained from the validation data. In addition, the update function could be implemented by the sum or concatenation operation. We set $K = 1$ for all the experiments.

The performance of BFReg-NN is influenced by the value of hyperparameter $\alpha$. While $\alpha$ controls the acceptance of newly discovered knowledge, it is dependent on whether the existing biological knowledge is suitable for the current dataset, as shown in Table 8. For example, in the future value forecasting task, $\alpha$ is larger than in other tasks to get the best result because existing knowledge is incomplete in the dataset. Besides, in the missing value task, our model performs better when $\alpha = 0$ in the protein level, which means BFReg-NN ignores the irrelevant level when discovering knowledge. We vary it from $\{0, 0.00001, 0.00005, 0.0001, 0.0005, 0.001, 0.005\}$ and select the best value when the loss is the smallest in the validation dataset.

## A.2    MODEL COMPLEXITY

The complexity of the BFREg-NN model depends on the number of levels $L$ and the number of biological factors at each level $n_l$. Therefore, the model complexity is $O(\sum_{l=1}^{L} n_l^2)$. In the real application, we choose the potentially important genes. If the computation resource is adequate, we could set as many genes as possible. In this paper, we use a single A100 GPU to run experiments.

# B    APPENDIX - OTHER BASELINES

We complement three baselines, Random Forest, XGboost and DCell, in this section.

## B.1 CELL CLASSIFICATION TASK

Table 9: other baselines for the cell classification task.

| Models | GSE | muscle | diaphragm | lung | trachea |
|---|---|---|---|---|---|
| Random Forest | 0.9688 | 0.8771 | 0.7161 | 0.8834 | 0.9273 |
| XGBoost | 0.8867 | 0.8534 | 0.7549 | 0.8625 | 0.9193 |
| DCell | 0.9482 | 0.7158 | 0.6731 | 0.7545 | 0.9322 |
| BFReg-NN(basic) | 0.9476 | 0.8798 | 0.8420 | 0.8808 | 0.9376 |
| BFReg-NN(enhanced) | **0.9693** | **0.8884** | **0.8509** | **0.8903** | **0.9446** |

Although in single-cell tasks, Random Forest and XGBoost models could have comparable performance, they are still worse than BFReg-NN on most of the datasets in Table 9.

DCell (Ma et al., 2018) used the Gene Ontology (GO) knowledgebase to build the neural network architecture, which limits the performance. GO defines GO terms (e.g., molecular function, cellular component, biological process) and builds the architecture based on term relations. Since each GO term includes several genes, the authors leverage the genes as the input layer and GO terms as the following layers in the neural network. We think the network is too sketchy to simulate the gene/protein reactions in the cell, which may lead to suboptimal performance. Instead, BFReg-NN uses more specific knowledge (gene regulatory network, protein-protein interactions and pathways) to mimic the cell system. Therefore, BFReg-NN(basic) is much better than DCell on the muscle/diaphragm/lung dataset. Further, BFReg-NN has the potential ability to discover new biological relations from inputs (i.e., gene expression data) to enhance performance. As shown in Table 9, the enhanced BFReg-NN achieved the best results.

## B.2 FUTURE GENE EXPRESSION PREDICTION

Table 10: other baselines for future gene expression task.

| Model | MSE | PCC |
|---|---|---|
| RF | $0.1115 \pm 0.0021$ | $0.9685 \pm 0.0006$ |
| XGBoost | $0.0837 \pm 0.0033$ | **0.9753**$\pm 0.0008$ |
| BFReg-NN (MLP,enhanced) | **0.0724**$\pm 0.0023$ | $0.9724 \pm 0.0007$ |

The best-performing models in the 2019 Dream Challenge are ensemble methods, which include several sub-models of Random Forest and XGBoost. Here we verify the performance of a single model shown in Table 10. In the future gene expression prediction task, Random Forest has pool performance on both MSE and PCC metrics. XGboost has a little higher value on PCC value but also a higher MSE value, which means XGboost could predict the trend of the cells but fail on the specific values. However, they lack the interpretation of the gene regulatory level from the biological view, and they can not discover new biological knowledge.

## C APPENDIX - PRE-TRAINING

Table 11: performances of pre-trained BFReg-NN.

| Model | MSE | PCC |
|---|---|---|
| BFReg-NN | $0.0724 \pm 0.0023$ | $0.9724 \pm 0.0007$ |
| Pre-trained BFReg-NN | **0.0719**$\pm 0.0042$ | $0.9723 \pm 0.0018$ |

BFReg-NN could benefit from the pre-training and fine-tuning framework. The detailed experimental steps are: (1) Utilize the missing gene expression prediction task to pre-train a BFReg-NN model on the breast cancer dataset; (2) Freeze the parameters of the BFReg-NN model except for the last

MLP layers, and (3) Fine-tune the last MLP layers to do the future gene expression prediction. The results are shown in Table 11.

With the pre-training on the missing value task, the performance increases significantly on the MSE metric. Because biological knowledge is universal and similar in the cell across different tasks, it is reasonable that pre-trained BFReg-NN improves the other tasks' performance with fine-tuning. However, the standard deviation slightly rises, which means a more stable method to fine-tune the model should be discussed in the future.

# D APPENDIX - ABLATION STUDIES

## D.1 TOP-20 GENE PAIRS FOR P.P53 GENE

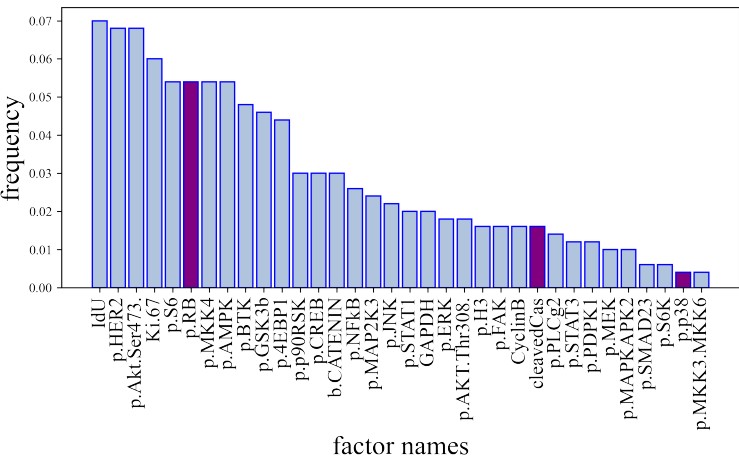

Figure 4: Top-20 frequent interactions for gene p.p53

Figure 4 shows the top-20 frequent interactions for gene p.p53 in breast cancer. The purple color marks the relations that have been identified in the existing knowledge. All the related genes of p.p53 are discovered by BFReg-NN. The recall of p.p38 is 0.4933 averaged by the total runs.

## D.2 TRAINING SAMPLE INFLUENCE

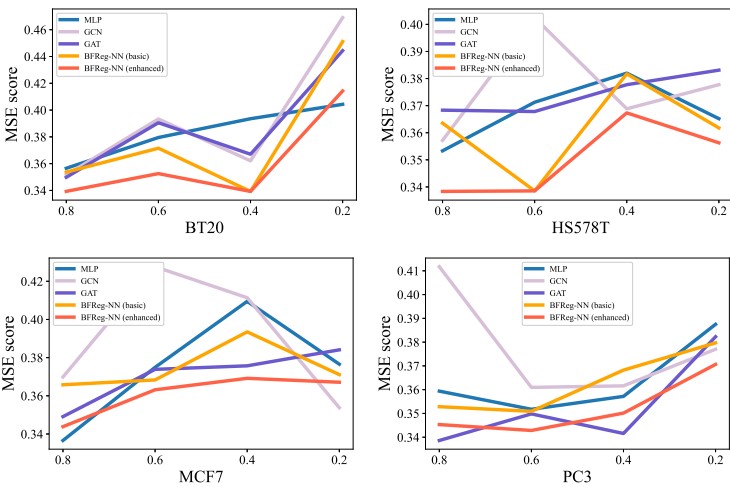

Figure 5: Influence on missing value task when reducing cell samples

We test the model performance with the number of training samples. Take the missing value task as an example, shown in Figure 5. We randomly leave the cell samples in {0.8, 0.6, 0.4, 0.2} proportions of the dataset, then run the models 5 times to compute the average MSE score. Most models perform badly when removing more samples. GCN is not stable with a steep curve. MLP performs worse than others because it only relies on gene expression data while others are also supported by existing biological knowledge. Compared to GAT, BFReg-NN (basic) and BFReg-NN (enhanced) have better results. Also, BFReg-NN (enhanced) has the lowest MSE scores and is less influenced by small cell samples. The reason is that BFReg-NN (enhanced) has the ability to build new knowledge as the additional features to predict the gene expression.

## D.3 STABILITY ANALYSIS

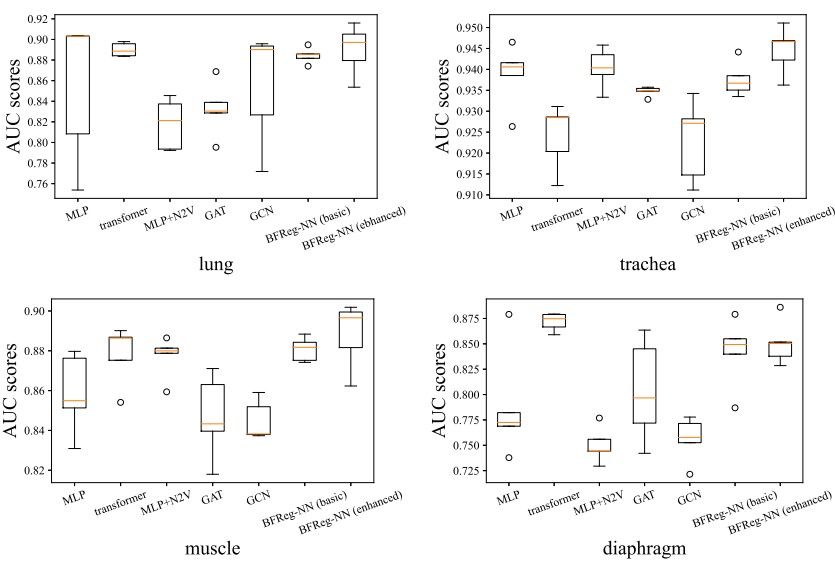

Figure 6: Distribution of AUC score on the cell classification task

In this section, we report the distribution of AUC score on the cell classification task, shown in Figure 6. We ignore the GatedNN and GCN/GAT with co-expression knowledge due to their poor performance. The results of MLP are in a large range or have outliers, indicating it is severely influenced by random seeds. GCN, transformer and BFReg-NN (enhanced) have fewer outliers over four datasets. GCN has a low AUC score, while transformer and BFReg-NN (enhanced) perform well in both stability and accuracy, which means learning complex relations is beneficial in the cell classification task. However, the average score of BFReg-NN (enhanced) is still higher than the transformer in most cases, because it uses confident knowledge to define important regulations. GAT and BFReg-NN (basic) are less stable than the above three, but BFReg-NN (basic) achieves better results than GAT. In conclusion, BFReg-NN (enhanced) consistently obtains stable and accurate predictions.

Table 12: Stablity analysis of BFReg-NN.

| Metrics | GSE | muscle | diaphragm | lung | trachea |
|---------|--------|--------|-----------|--------|---------|
| AUC | 0.9399 | 0.9046 | 0.8201 | 0.9065 | 0.9540 |
| STD | 0.0583 | 0.0096 | 0.0213 | 0.0193 | 0.0059 |

More specifically, we randomly split the whole dataset into training/validation/test sets and verify BFReg-NN model stability in Table 12. We observe that the standard deviations are in a small value, which means BFReg-NN is stable.

