# OpenReview forum: "Biological Factor Regulatory Neural Network"
_ICLR.cc/2023/Conference — Submitted to ICLR 2023_

### Official Review · Reviewer_1ymJ · 2022-10-21

**Confidence:** 4
**Correctness:** 3
**Technical Novelty And Significance:** 2
**Empirical Novelty And Significance:** 3
**Recommendation:** 3

**Clarity, Quality, Novelty And Reproducibility:**

In general, I found the text very hard to understand, despite being familiar with this area. Many terms not explicitly defined, such as "biological factor", "level", "[neural network] hyperedge", "knowledge". The text includes a great deal of imprecise language such as "almost transparent", "as much as possible", "most typical"

**Strength And Weaknesses:**

Many approaches have been developed that aim to incorporate biological databases into neural network architecture. For example, see section "Using prior knowledge for transparent models" of the review below, which includes some missing citations. I couldn't tell which parts of the proposed strategy are novel. Also, how did the authors choose which existing methods to compare to?
https://www.nature.com/articles/s41576-022-00532-2

What are the state-of-the-art methods for each of the three target tasks? How does BFReg-NN's compare to these methods? It's okay if BFReg-NN has worse performance in the name of interpretability, but it is important at least to know what the loss is.

5.4: It is impossible to judge whether 0.4172 is a good recall without a measure of precision. Is BFREG-NN predicting many non-edges. One way to evaluate this would be to rank potential edges by how highly they were predicted, then show how high the known edges are in this ranking. Even if there are some previously-unknown true edges, a good predictor would rank known edges highly.

Minor notes:

Both parts of this statement are too strong: "It is easy for DNNs to diagnose cancer for patients, but DNNs cannot tell us how biological processes cause cancers"

I don't understand this: "Moreover, regulations and reactions happen at both intra-level and inter-level modeled in the cell system view. Graph neural network styled operations for intra-level (hyper)edges and deep neural network styled operations for inter-level mappings, are applied respectively."

Ref "Yao et al., 2030"
Incorrectly cited: "Consortium et al. (2018)".


**Summary Of The Paper:**

The authors present a neural network method BFReg-NN for several tasks involving RNA-seq data. This method defines its architecture using biological databases that annotate gene regulatory networks, protein-protein interactions and pathways, represented as graphs/hypergraphs. The neural network uses graph attention to perform computations on these graphs. The authors apply BFReg-NN to several tasks: imputing held-out gene expression values, predicting cell type, and predicting future gene expression values.

**Summary Of The Review:**

The problem of using prior knowledge to design transparent NN architectures is important and the authors' approach seems reasonable. However, there is a great deal of literature on this topic already and I could not tell which parts of the proposed approach are novel.

---

> ### Author Response · Authors · 2022-11-18
> **Reviewer 1ymJ part 3**
>
> > Q: It is impossible to judge whether 0.4172 is a good recall without a measure of precision. Is BFREG-NN predicting many non-edges. One way to evaluate this would be to rank potential edges by how highly they were predicted, then show how high the known edges are in this ranking. Even if there are some previously-unknown true edges, a good predictor would rank known edges highly.
>
>
> Figure address: https://www.dropbox.com/s/fmjtahm21r48oa4/fig_gene.png?dl=0
>
> A: Thanks for your question. Here we take an example of gene p.p53 to show how well we complete the existing knowledge. We remove its related edges to make it an isolated node, and then conduct the model to recover the knowledge. BFReg-NN is run several times to avoid randomness. In each run, the discovered edges come out with weights, and are ranked to select the top-20 list. The frequency of an edge is computed by (the times that the edge is in the top-20 list)/(the total runs). The above figure shows the result. We can find all the existing edges (p.RB, cleavedCas, p.p38) marked by purple for gene p.p53 in the top-20 list. The recall of p.p38 is 0.4933 averaged by the total runs. We have added this example in the revision.
>
>
> > Q: Both parts of this statement are too strong: "It is easy for DNNs to diagnose cancer for patients, but DNNs cannot tell us how biological processes cause cancers"
>
> A: Thanks for your question. We have modified it to: "Although DNNs model could diagnose cancers with high precision, the original DNNs cannot tell us detailed biological factors/processes which cause cancers. " We have edited it in the revision.
>
> > Q: I don't understand this: "Moreover, regulations and reactions happen at both intra-level and inter-level modeled in the cell system view. Graph neural network styled operations for intra-level (hyper)edges and deep neural network styled operations for inter-level mappings, are applied respectively."
>
> A: Thanks for your question. We have modified it to: "Moreover, two different operations are utilized to simulate the reactions inside/across the layer. Since genes regulate each other and create feedback loops to form cyclic chains of dependencies in gene regulatory network, graph neural network styled operations are suitable to model the "steady states" of genes. It is the same for proteins in PPI. In the layer of pathways, it is a hypergraph where each hyperedge is a pathway including multiple proteins. Accordingly, a hypergraph neural network is used to aggregate and balance the information of each protein. The operations across the layers imitate the material transformation (e.g., genes translate protein), so we adopt deep neural network styled operations to map." We have edited it in the revision.
>
>
> > Q: Ref "Yao et al., 2030" Incorrectly cited: "Consortium et al. (2018)".
>
> A: Sorry for the wrong citation. We have edited it.
>
> > Q: In general, I found the text very hard to understand, despite being familiar with this area. Many terms not explicitly defined, such as "biological factor", "level", "[neural network] hyperedge", "knowledge". The text includes a great deal of imprecise language such as "almost transparent", "as much as possible", "most typical"
>
> A: Sorry for the unclear descriptions and we have refined them in our reversion. "Biological factor" means the different entities or entity sets in the cells defined by human beings. Entities include biological molecules, such as genes and proteins. Entity sets include pathways and biological processes. "Level" is a term in system biology, where each level is associated with a type of biological factor. "Hyperedge" is a special edge in a hypergraph, indicating that an edge could connect more than two nodes. "Knowledge" in this paper means the relations of biological factors, which are already found by biological experiments. In our work, the knowledge can also be discovered by enhanced BFReg-NN in an edge form. As for imprecise language, we also edited the paper.

---

> > ### Comment · Reviewer_1ymJ · 2022-11-20
> > **2022-11-20**
> >
> > Thank you for the response. These experiments significantly improve the paper.
> >
> > > The best-performing models are ensembling several Random Forest and XGBoost claimed in Gabo et al. (2021).
> >
> > It is true that some of the best-performing teams used RF and XGBoost, but there was substantially more to those methods than just plugging in off-the-shelf models. The authors need to compare to the actual approaches proposed by the teams.
> >
> > Unfortunately, it is a bit hard to evaluate a paper that would need to be substantially rewritten to take into account these new experiments and to improve its clarity. If this paper is not accepted now, I think it has a good chance in a future submission, when the reviewers can see the improvements. I update my score to 5.

---

> > > ### Author Response · Authors · 2022-11-21
> > > **Reviewer 1ymJ part 4**
> > >
> > > Thanks for your comment. We would appreciate it if you could update your score in the system. It is a rush time for us to complete all experiments. We will include more specific experiments in the final version.

---

> ### Author Response · Authors · 2022-11-18
> **Reviewer 1ymJ part 2**
>
> > Q: What are the state-of-the-art methods for each of the three target tasks? How does BFReg-NN's compare to these methods? It's okay if BFReg-NN has worse performance in the name of interpretability, but it is important at least to know what the loss is.
>
> A: The best-performing models are ensembling several Random Forest and XGBoost claimed in Gabo et al. (2021). We also include Ma et al.(2018) for comparison in the cell classification task. Here we test the single model performance running 5 times and calculate the average scores shown below:
>
> Although in single-cell tasks, Random Forest and XGBoost models could have comparable performance, they are still weaker than BFReg-NN model. Besides, they can not interpret the molecule factor interactions.
>
>
> **Table: Results in cell classification task**
> |                    | GSE     | muscle  | diaphragm | lung    | trachea |
> |--------------------|---------|---------|-----------|---------|---------|
> | RF                 | 0.9688  | 0.8771  | 0.7161    | 0.8834  | 0.9273  |
> | XGBOOST            | 0.8867  | 0.8534  | 0.7549    | 0.8625  | 0.9193  |
> | DCell            | 0.9482  | 0.7158  | 0.6731    | 0.7545  | 0.9322  |
> | BFReg-NN(enhanced) | **0.9693**  | **0.8884**  | **0.8509**    | **0.8903**  | **0.9446**  |
>
>
>
> **Table: Results in future gene expression forecasting task**
> | Model                   | MSE     | PCC     |
> |-------------------------|---------|---------|
> | RF                      | 0.1115 $\pm$ 0.0021 | 0.9685 $\pm$ 0.0006|
> | XGBOOST                | 0.0837 $\pm$ 0.0033| **0.9753** $\pm$ 0.0008|
> | BFReg-NN (MLP,enhanced) | **0.0724** $\pm$ 0.0023 | 0.9724 $\pm$ 0.0007|
>
>
> Gabo et al. (2021) Gabor A, Tognetti M, Driessen A, et al. Cell‐to‐cell and type‐to‐type heterogeneity of signaling networks: insights from the crowd[J]. Molecular systems biology, 2021, 17(10): e10402.

---

> ### Author Response · Authors · 2022-11-18
> **Reviewer 1ymJ part 1**
>
> > Q: Many approaches have been developed that aim to incorporate biological databases into neural network architecture. For example, see section "Using prior knowledge for transparent models" of the review below, which includes some missing citations. I couldn't tell which parts of the proposed strategy are novel. Also, how did the authors choose which existing methods to compare to? https://www.nature.com/articles/s41576-022-00532-2
>
> A: Our model belongs to "using prior knowledge for transparent models" according to Gherman et al. (2022). Compared with the existing methods, BFReg-NN has significant differences summarized in two points.
>
> First of all, we take DCell as a typical models of "using prior knowledge for transparent models". DCell use the Gene Ontology (GO) knowledgebase to build the neural network architecture, which limits the performance. GO defines GO terms (e.g., molecular function, cellular component, biological process) and builds the architecture based on term relations. Since each GO term includes several genes, the authors leverage the genes as the input layer and GO terms as the following layers in the neural network. We think the network is too sketchy to simulate the gene/protein reactions in the cell, which may lead to suboptimal performance. Thus, we proposed BFReg-NN, which uses more specific knowledge (gene regulatory network, protein-protein interactions and pathways) to mimic the cell system.
>
> Secondly, compared with the transparent models, P-Net and DCell, BFReg-NN employs hypergraphs to represent pathways/biological processes. The hypergraphs allow the genes and proteins to aggregate the information directly and thus get more adaptive representations for a biological phenomenon. However, the existing work just merged them together into a pathway or biological process unit. Thus, BFReg-NN has a better performance than the existing works, especially compared with DCell and P-Net.
>
> We added the experiments to implement the model DCell from Ma et al.(2018) for the cell classification task, and run 5 times to show the averaged result in the following table. BFReg-NN(basic) is much better than DCell on the muscle/diaphragm/lung dataset. We have also cited these papers in the revision.
>
> | Models             | GSE     | muscle  | diaphragm | lung    | trachea |
> |--------------------|---------|---------|-----------|---------|---------|
> | DCell             | 0.9482  | 0.7158  | 0.6731    | 0.7545  | 0.9322  |
> | BFReg-NN(basic)    | 0.9476  | 0.8798  | 0.8420    | 0.8808  | 0.9376  |

---

### Official Review · Reviewer_GToB · 2022-10-21

**Confidence:** 4
**Clarity, Quality, Novelty And Reproducibility:** See above.
**Correctness:** 3
**Technical Novelty And Significance:** 3
**Empirical Novelty And Significance:** 3
**Recommendation:** 6

**Strength And Weaknesses:**

Strengths: 1. This paper which takes different levels of biological prior information and formulates them in a hierarchical structure for potential discoveries is novel.
2. The experiments are comprehensive.

Weaknesses: 1. There is no theoretical guarantee that the discoveries resulting from the sparse network architecture are unique.
                      2. The paper missed the description of how to quantify the top-k frequent interaction pairs, which is discussed in section 5.4. As the paper claims, one advantage of this framework is novel discovery. However,  very limited results and discussions are presented here.
                      3. It lacks model complexity analysis and comparison. Given different levels of biological entities' intra and inter interactions, I am worried about the model's real applicability.




**Summary Of The Paper:**

This paper proposes a framework based on some existing works to incorporate gene regulatory networks (Garcia-Alonso et al., 2019), protein-protein interaction (Chereda et al., 2021) networks, protein-pathway networks (Elmarakeby et al., 2021), and relations among genes, proteins, and pathways for biological discoveries. The idea is very interesting, and the empirical results showed its superiority.

**Summary Of The Review:**

Overall, I like the paper's idea, and it is new to me to have so many biological layers of information in one model for downstream analysis. However, I have some concerns regarding the sparse architecture and the interpretation of this model.

1:  PNET builds the sparse network by the sparse connections between genes and pathways from the Reactome database. How did this happen for your sparse connection between gene-mRNA, mRNA-Protein, and protein-Pathway?

2:  Compare to PNET, they have the ranking of top features in different layers which is meaningful for biomarker discovery, can BFReg-NN do this?

3:  Why does the paper put an embedding layer to encode each gene rather than set each node as one gene, mRNA, or protein? What are the intuitions and benefits of the embedding layer?

4:  The enhanced BFREG-NN model is not clear to me. How did it transform $A_l$ to $A_l^`$?

---

> ### Author Response · Authors · 2022-11-18
> **Reviewer GToB, part 3**
>
> > Q: Why does the paper put an embedding layer to encode each gene rather than set each node as one gene, mRNA, or protein? What are the intuitions and benefits of the embedding layer?
>
> A: The embedding layer is a MLP with the same parameters for each gene. The input of the embedding layer is a float number representing the count of a gene, and the output is a high-dimension vector of that gene. As the gene expression is in a float form with possible small measurement errors, the MLP is utilized to reduce this error and thus enhance the performance of the model.
>
> > Q: The enhanced BFREG-NN model is not clear to me. How did it transform $A_l$ to $A_l'$
>
> A: Thanks for your question. We define biological knowledge to be discovered and undiscovered. Discovered knowledge is in a binary matrix $\mathbf{A}_l$, deciding the message passing among neurons. In the enhanced BFReg-NN, we learn the weighted matrix $\mathbf{A}_l'$ under the constraint of $\mathbf{A}_l$, which includes the possibility of undiscovered edges. We have edited the descriptions more clearly in Sec 4.3 as "Instead of a binary matrix $\mathbf{A}_l$ used in the basic model, we use $\mathbf{A}_l$ to constrain the learnable matrix $\mathbf{A'}_l$ to discover new knowledge."

---

> > ### Comment · Reviewer_GToB · 2022-12-04
> > **Response to authors**
> >
> > Thank you for the explanations. I will keep my rate.

---

> ### Author Response · Authors · 2022-11-18
> **Reviewer GToB, part 2**
>
> > Q: It lacks model complexity analysis and comparison. Given different levels of biological entities' intra and inter interactions, I am worried about the model's real applicability.
>
> A: The complexity of the BFREg-NN model depends on the number of levels $L$ and the number of biological factors at each level $n_l$. Therefore, the model complexity is $O(\sum^L_{l=1}{n_l}^2)$. In the real application, we choose the potentially important genes. If the computation resource is adequate, we could set as many genes as possible. In this paper, we use a single A100 GPU to run experiments.
>
>
> > Q: PNET builds the sparse network by the sparse connections between genes and pathways from the Reactome database. How did this happen for your sparse connection between gene-mRNA, mRNA-Protein, and protein-Pathway?
>
> A: BFReg-NN is flexible. If the prior knowledge only includes genes and pathways from the Reactome database, we could remove the protein layers.
>
> > Q: Compare to PNET, they have the ranking of top features in different layers which is meaningful for biomarker discovery, can BFReg-NN do this?
>
> A: Thanks for the insightful question. Actually, this is also an advantage of BFReg-NN. Specifically, the important genes or proteins could be ranked by the weighted matrix $\mathbf{W}^l$ in equation 2. We further highlight it in Sec 4.3.

---

> ### Author Response · Authors · 2022-11-18
> **Reviewer GToB, part 1**
>
> > Q: There is no theoretical guarantee that the discoveries resulting from the sparse network architecture are unique.
>
> A: Thanks for your advice. Currently, there is no theoretical guarantee but empirical experiments in the paper. For the reported results, we run BFReg-NN multiple times to guarantee the discoveries are not random.
>
> > Q: The paper missed the description of how to quantify the top-k frequent interaction pairs, which is discussed in section 5.4. As the paper claims, one advantage of this framework is novel discovery. However, very limited results and discussions are presented here.
>
> A: Sorry for the unclear descriptions. We count the discovered edges by running BFReg-NN several times to ensure the high possibility of potential edges. In each run, the discovering edges would come out with weights. We sort the edges by their weights and select the top-k edges. The frequency of an edge is computed by (the times that the edge is in the top-k list)/(the total runs). In this paper, we take gene regulatory discovery as an example, which is the basic part of research in biological mechanisms. The more discovered knowledge, such as the new gene or protein interactions, should be verified by biological methods and beyond the discussion of the paper.

---

### Official Review · Reviewer_PD4L · 2022-10-23

**Confidence:** 3
**Correctness:** 3
**Technical Novelty And Significance:** 1
**Empirical Novelty And Significance:** Not applicable
**Recommendation:** 3

**Clarity, Quality, Novelty And Reproducibility:**

The presentation of the paper can be improved.
* problem definition in sec 3.2 is unclear; It reads more like the high-level description of the model.
* It would be helpful to elaborate on how “pathways” is designed. Some specific questions I have when reading the paper:
1. What are the edges between pathways?
 2. how can Enrichr represent pathways? It is a collection of genesets.
 3. What’s the reason for using LINCS genesets for predicting L1000 gene expression? The ligand perturbation genesets are not “pathways.” And would it introduce any leak by using such genesets?
 4. I am unsure what graphs are used when forecasting gene expression time series.


**Strength And Weaknesses:**

Regarding novelty
Ingesting prior knowledge into neural network architecture is not a new idea. A few studies have studied the problem of Ingesting prior knowledge into neural network architecture for explainability, but those studies are not discussed in the paper.
Yu MK, Kramer M, Dutkowski J, Srivas R, Licon K, Kreisberg JF, Ng CT, Krogan N, Sharan R, Ideker T. Translation of genotype to phenotype by a hierarchy of cell subsystems. Cell systems. 2016 Feb 24;2(2):77-88.
Ma J, Yu MK, Fong S, Ono K, Sage E, Demchak B, Sharan R, Ideker T. Using deep learning to model the hierarchical structure and function of a cell. Nature methods. 2018 Apr;15(4):290-8.

Regarding evaluation:
It is confusing since the L1000 dataset used in the missing gene expression prediction task is not collected by single-cell sequencing, while section 4.4 motivates the task by mentioning the problem of dropout events in single-cell sequencing. For L1000, predicting the expression of non-landmark genes might be a useful task for this model.
It may be worth discussing the best-performing models from the 2019 Dream Challenge for performance comparison.

The technical depth of this paper seems to be limited, in my opinion.


**Summary Of The Paper:**

The paper presents a neural network model whose weights and connections between nodes are derived from prior biological knowledge. Every node in the neural network represents a biological entity such as a gene, a protein, a pathway, or a phenotype. Besides injecting prior knowledge into neural network architecture, the paper also presents a mechanism to learn weights for non-existence edges to complete the prior knowledge. The model's performance is evaluated by three tasks: imputing missing genes, classifying cell types, and forecasting gene expression over time.

**Summary Of The Review:**

I wouldn’t recommend this paper for ICLR. It is a decent paper, but given the lack of technical depth and novelty, it seems other forums focusing on biology or bioinformatics would fit this paper better.

---

> ### Author Response · Authors · 2022-11-18
> **Reviewer PD4L, part 3**
>
> > Q: problem definition in sec 3.2 is unclear; It reads more like the high-level description of the model.
>
> A: Thanks for your question. We have moved Sec 3.2 to Sec 4 and only left the Biological background in Sec 3 for more clear descriptions.
>
>
> > Q: It would be helpful to elaborate on how “pathways” is designed. Some specific questions I have when reading the paper:
> What are the edges between pathways?
> how can Enrichr represent pathways? It is a collection of genesets.
>
> A: In the cell system, pathways mean a collection of genes to do a specific biological activity. In BFReg-NN, we model the layer of pathways by a hypergraph. The edge in the hypergraph is hyperedge, which could connect more than two nodes (proteins). The hypergraph neural network allows the nodes in the same hyperedge to propagate and aggregate the information directly. Thus, it can get more adaptive representations for a biological phenomenon.
>
> Enrichr is a dataset for pathways, indicating that the potential biological activities consist of different genes. We employ the knowledge from Enrichr to build the hypergraph to represent the biological activities in a cell.
>
> > Q: What’s the reason for using LINCS genesets for predicting L1000 gene expression? The ligand perturbation genesets are not “pathways.” And would it introduce any leak by using such genesets?
>
> A: The LINCS genesets are as far as we can reach to build the pathway, which is the most relative to the L1000 dataset. Despite that, the knowledge set is easy to change. It is not limited to LINCS genesets. As the genesets in pathways are knowledge for the hypergraph, they do not leak the information for the downstream tasks.
>
> > Q: I am unsure what graphs are used when forecasting gene expression time series.
>
> A: We used the graphs provided by the 2019 Dream Challenge. It is a small protein-protein interactions graph where nodes are phosphorylated genes.

---

> > ### Comment · Reviewer_PD4L · 2022-11-28
> > **Thanks for the detailed response**
> >
> > Thanks for the detailed response. I share reviewer 1ymJ's concern, and I will keep my rate.

---

> ### Author Response · Authors · 2022-11-18
> **Reviewer PD4L, part 2**
>
> > Q: Regarding evaluation: It is confusing since the L1000 dataset used in the missing gene expression prediction task is not collected by single-cell sequencing, while section 4.4 motivates the task by mentioning the problem of dropout events in single-cell sequencing. For L1000, predicting the expression of non-landmark genes might be a useful task for this model
>
> A: Thanks for your question. Because of the limitation of sequencing technology, the L1000 dataset does not have the golden standard values for non-landmark genes. However, we need to compare the model performance, so we assume the processed L1000 dataset as the golden standard, and thus we can fairly compare the model performance.
>
> > Q: It may be worth discussing the best-performing models from the 2019 Dream Challenge for performance comparison.
>
> A: Thanks for your advice. Our submission version only focuses on the deep learning methods, which have the potential abilities to model the biological factor connections. The best-performing models in the 2019 Dream Challenge are ensemble methods, which include several sub-models of Random Forest and XGBoost. Here we verify the performance of a single model below:
>
> | Model                   | MSE     | PCC     |
> |-------------------------|---------|---------|
> | Random Forest           | 0.1115 $\pm$ 0.0021 | 0.9685 $\pm$ 0.0006|
> | XGBoost                 | 0.0837 $\pm$ 0.0033| **0.9753** $\pm$ 0.0008|
> | BFReg-NN (MLP, enhanced) | **0.0724** $\pm$ 0.0023 | 0.9724 $\pm$ 0.0007|
>
> In the future gene expression prediction task, Random Forest has pool performance on both MSE and PCC metrics. XGboost has a little higher value on PCC value but also a higher MSE value, which means XGboost could predict the trend of the cells but fail on the specific values. However, they lack the interpretation from the gene regulatory level in the biological view, and they can not discover new biological knowledge.

---

> ### Author Response · Authors · 2022-11-18
> **Reviewer PD4L, part 1**
>
> > Q: Regarding novelty Ingesting prior knowledge into neural network architecture is not a new idea. A few studies have studied the problem of Ingesting prior knowledge into neural network architecture for explainability, but those studies are not discussed in the paper. Yu MK, Kramer M, Dutkowski J, Srivas R, Licon K, Kreisberg JF, Ng CT, Krogan N, Sharan R, Ideker T. Translation of genotype to phenotype by a hierarchy of cell subsystems. Cell systems. 2016 Feb 24;2(2):77-88. Ma J, Yu MK, Fong S, Ono K, Sage E, Demchak B, Sharan R, Ideker T. Using deep learning to model the hierarchical structure and function of a cell. Nature methods. 2018 Apr;15(4):290-8.
>
> A: Thanks for providing two related papers. We cited these two papers and discussed them in our revised version. However, there are two major differences between our proposed BFReg-NN and their works. They use partial biological knowledge, as we have claimed in the submission version: "Although they mitigate the black-box issues, they only use partial biological knowledge, and they cannot explore new knowledge from gene expression data."
>
> First of all, these papers use the Gene Ontology (GO) knowledgebase to build the neural network architecture, which limits the performance. GO defines GO terms (e.g., molecular function, cellular component, biological process) and builds the architecture based on term relations. Since each GO term includes several genes, the authors leverage the genes as the input layer and GO terms as the following layers in the neural network. We think the network is too sketchy to simulate the gene/protein reactions in the cell, which may lead to suboptimal performance. Thus, we proposed BFReg-NN, which uses more specific knowledge (gene regulatory network, protein-protein interactions and pathways) to mimic the cell system. We added the experiments to implement the model from Ma et al.(2018) for the cell classification task, and run 5 times to show the averaged result in the following table. BFReg-NN(basic) is much better than DCell on the muscle/diaphragm/lung dataset.
>
> Secondly, these two pieces of work do not consider discovering unknown interactions. Our model has the potential ability to discover new biological relations from inputs (i.e., gene expression data) to enhance the performance further. As shown in the table, the enhanced BFReg-NN achieved the best results.
>
> | Models             | GSE     | muscle  | diaphragm | lung    | trachea |
> |--------------------|---------|---------|-----------|---------|---------|
> | DCell             | 0.9482  | 0.7158  | 0.6731    | 0.7545  | 0.9322  |
> | BFReg-NN(basic)    | 0.9476  | 0.8798  | 0.8420    | 0.8808  | 0.9376  |
> | BFReg-NN(enhanced) | 0.9693  | 0.8884  | 0.8509    | 0.8903  | 0.9446  |

---

### Official Review · Reviewer_J4we · 2022-10-24

**Confidence:** 4
**Clarity, Quality, Novelty And Reproducibility:** Authors need to clarify some details …
**Correctness:** 4
**Technical Novelty And Significance:** 2
**Empirical Novelty And Significance:** 2
**Recommendation:** 5

**Strength And Weaknesses:**

It’s an important question to understand the black box characteristics of neural networks given its superior performance. Authors addressed this question leveraging the existing knowledge to design biologically meaningful architectures for BFReg-NN. The manuscript is well written with a comprehensive review of the field. There are some experiment details that need to be clarified as follows.

Major Questions

1.	(Section 4.2) Could you elaborate the embedding layer that was utilized to embed each gene?

2.	(Section 5.2, Table 2-4) Could you repeat the experiments multiple times varying the training/validation/test split and report the standard deviations to justify the performance? This information is likely to be available because of the discussion in the section A.2 stability analysis.

3.	(Section 5.3) As Table 3 and Table 5 are both related to the task of cell classification. The performance of predicting muscle (Basic AUC = 0.8798) in Table 3 corresponds to the GRN&PPI according to Table 5. The performance of predicting diaphragm (Basic AUC = 0.8420) corresponds to GRN&PPI&Pathway based on Table 5. It seems the performances reported in Table 3 are corresponding to different combinations of knowledge. Could you elaborate on this?

4.	(Table 8) alpha was tuned via grid search based on the appendix. It’s not clear if it was tuned based on the performance on the validation set. Also did alpha vary in the ablation test?

5.	(Discussion) For each of three tasks presented in the paper, authors trained BFReg-NN separately. If BFReg-NN learns the underlying biological parameters through training, it may have a good performance in a related task even after being trained with respect to a different task. For example, did the parameters learned with respect to the task of missing gene expression value prediction also perform well in the experiment of future gene expression value forecasting with limited fine tuning? If there are cell lines shared across these two tasks, this may be a valid experiment to pursue.

Minor Questions

1.	BFReg-NN is designed to mimic the hierarchical biological network in nature. Is there a way to incorporate the feedback loop across hierarchies to make RFReg-NN architecture more similar to the real biological regulatory hierarchy. This is definitely beyond the scope of this manuscript, but a natural extension to consider.



**Summary Of The Paper:**

Authors proposed BFReg-NN as a general deep learning model and designed its architecture based on the regulatory relations and hierarchical relations among genes, proteins and pathways. Incorporating the biological knowledge into the network architecture design, authors tried to break the “black-box” nature of neural networks and learn new biologically insights from the data. BFReg-NN achieved superior performance in three gene expression-based tasks compared with the baseline. Authors also conducted ablation tests to highlight the contribution of each module in BFReg-NN. 

**Summary Of The Review:**

The manuscript is not ready yet to be presented in its current form. Extra work needs to be done to polish the manuscript. 

---

> ### Author Response · Authors · 2022-11-18
> **Reviewer J4we, part 3**
>
> > Q: BFReg-NN is designed to mimic the hierarchical biological network in nature. Is there a way to incorporate the feedback loop across hierarchies to make RFReg-NN architecture more similar to the real biological regulatory hierarchy. This is definitely beyond the scope of this manuscript, but a natural extension to consider.
>
> A: Thanks for your suggestion. Our current version is a basic framework to model the biological network according to the central dogma of molecular biology. There are some special cases, such as the feedback loop of RNA replication, and reverse transcription of retroviruses. If the loop exists in one layer (e.g., RNA replication), BFReg-NN could conduct multiple GNN-based propagations and aggregations to simulate the finite loops. For reverse transcription of retroviruses, we can design a special network that allows the transfer of information from RNA to DNA, where the hierarchy is DNA->RNA->DNA->RNA->protein->downstream tasks.

---

> > ### Comment · Reviewer_J4we · 2022-12-11
> > **Response to the authors**
> >
> > I appreciate your detailed response. Most of my concerns have been addressed properly. There are two suggestions as a follow up.
> > 1. Please consider including part 2 in the future version to demonstrate BFReg-NN can learn the underlying biology regardless of the prediction tasks.
> > 2. One missing point is that there is no validation of new discovered edges regarding the enhanced BFReg-NN model. You can hold out some known connections in the interaction graph and see if the enhanced model can recover them.
> >
> > I will keep my scores at this point.

---

> ### Author Response · Authors · 2022-11-18
> **Reviewer J4we, part 2**
>
> >Q: (Discussion) For each of three tasks presented in the paper, authors trained BFReg-NN separately. If BFReg-NN learns the underlying biological parameters through training, it may have a good performance in a related task even after being trained with respect to a different task. For example, did the parameters learned with respect to the task of missing gene expression value prediction also perform well in the experiment of future gene expression value forecasting with limited fine tuning? If there are cell lines shared across these two tasks, this may be a valid experiment to pursue.
>
> A: Thanks for your valuable advice! We followed your suggestion to conduct the experiments and demonstrated that BFReg-NN can benefit from the pre-training and fine-tuning framework. The detailed experimental steps are: (1) Utilize the missing gene expression prediction task to pre-train a BFReg-NN model on the breast cancer dataset; (2) Freeze the parameters of the BFReg-NN model except for the last MLP layers, and (3) Fine-tune the last MLP layers to do the future gene expression prediction. The results are shown as:
>
> | Model                | MSE     | PCC     |
> |----------------------|---------|---------|
> | BFReg-NN              | 0.0724 $\pm$ 0.0023 | 0.9724 $\pm$ 0.0007|
> | BFReg-NN with pre-training| 0.0719 $\pm$ 0.0042 | 0.9723 $\pm$ 0.0018 |
>
> With the pre-training on the missing value task, the performance increases significantly on the MSE metric. Because biological knowledge is universal and similar in the cell across different tasks, it is reasonable that pre-trained BFReg-NN improves the other tasks' performance with fine-tuning. However, the standard deviation slightly rises, which means a more stable method to fine-tune the model should be discussed in the future.

---

> ### Author Response · Authors · 2022-11-18
> **Reviewer J4we, part1**
>
> > Q: (Section 4.2) Could you elaborate the embedding layer that was utilized to embed each gene?
>
> A: The embedding layer is a MLP with the same parameters for each gene. The input of the embedding layer is a float number representing the count of a gene, and the output is a high-dimension vector of that gene. As the gene expression is in a float form with possible small measurement errors, the MLP is utilized to reduce this error and thus enhance the performance of the model.
>
> > Q: (Section 5.2, Table 2-4) Could you repeat the experiments multiple times varying the training/validation/test split and report the standard deviations to justify the performance? This information is likely to be available because of the discussion in the section A.2 stability analysis.
>
> A: Thanks for your advice. We randomly split the data to training/validation/test 5 times in the cell classification task and report the results as follows:
>
> | Metrics | GSE    | muscle | diaphragm | lung   | trachea |
> |---------|--------|--------|-----------|--------|---------|
> | AUC     | 0.9399 | 0.9046 | 0.8201    | 0.9065 | 0.9540  |
> | STD     | 0.0583 | 0.0096 | 0.0213    | 0.0193 | 0.0059  |
>
> We observe that the standard deviations are in a small value, which means BFReg-NN is stable. Following your suggestion, we have added the results in Appendix A.3 in our revision.
>
> > Q: (Section 5.3) As Table 3 and Table 5 are both related to the task of cell classification. The performance of predicting muscle (Basic AUC = 0.8798) in Table 3 corresponds to the GRN&PPI according to Table 5. The performance of predicting diaphragm (Basic AUC = 0.8420) corresponds to GRN&PPI&Pathway based on Table 5. It seems the performances reported in Table 3 are corresponding to different combinations of knowledge. Could you elaborate on this?
>
> A: Sorry for the mistake we put the wrong value in Table 3. Basic AUC = 0.8798 in muscle should be changed to 0.8808.
>
> > Q: (Table 8) alpha was tuned via grid search based on the appendix. It’s not clear if it was tuned based on the performance on the validation set. Also did alpha vary in the ablation test?
>
> A: Thanks for your question. We not only use the validation data to decide the learned parameters in the model, but also tune the hyperparameter alpha. In detail, alpha is determined when the loss of validation data is the smallest. As for the ablation test, we study whether we use GRN, PPI or pathways in a special task. Alpha is different in each layer (GRN, PPI and pathways), so it needs to grid search and tuning by validation data. We have clarified them in Appendix A.1.

---

### Decision · Program_Chairs · 2023-01-20

**Decision:**

Reject

**Justification For Why Not Higher Score:**

N/A

**Justification For Why Not Lower Score:**

Most reviewers believe the paper is not ready for acceptance at this time. The authors need to better position their work by discussing other techniques for integrating biological knowledge with deep learning and contrasting with them. They also have to improve their experimental evaluation and compare with state of the art techniques, including black-box ones.  Finally, the authors should improve the writing to make the paper clearer and more accessible to a ML audience with limited background in bioinformatics.

**Metareview: Summary, Strengths And Weaknesses:**

The paper proposed a deep learning model for simulating biological processes. The main features of the proposed technique is the ability to integrate biological knowledge into the model, and the explainability of the model.

While the reviewers appreciate the contributions of the paper, they believe the paper is not yet ready for publication. The authors need to better position their work by discussing other techniques for integrating biological knowledge with deep learning and contrasting with them. They also have to improve their experimental evaluation and compare with state of the art techniques, including black-box ones.  Finally, the authors should improve the writing to make the paper clearer and more accessible to a ML audience with limited background in bioinformatics.  While the authors did try to address these issues during the discussion period, the changes are extensive enough to require a fresh review of the paper.